# Conducting Ethical Field Research on Rape in West African Settings: Case Study of 2018 Liberian Field Survey

**DOI:** 10.3390/healthcare11233053

**Published:** 2023-11-28

**Authors:** Jessi Hanson-DeFusco, Ernest Garnak Smith, Richard Fotorma Ngafuan, William N. Dunn

**Affiliations:** 1School of Economics, Political, and Policy Sciences (EPPS), University of Texas at Dallas, Richardson, TX 75248, USA; 2Department of Sociology, AME Zion University, Monrovia 1000, Liberia; esmithjr2@my.gcu.edu (E.G.S.J.); richardngafuan@live.com (R.F.N.); 3Graduate School of Public and International Affairs, University of Pittsburgh, Pittsburgh, PA 15206, USA; dunn@pitt.edu

**Keywords:** ethical research, Africa, sexual abuse, rape, contextuality, culturally sensitive, methodology, survey design, collaboration

## Abstract

Background: Rape scholarship in West Africa is growing, but studies often utilize Westernized approaches. A 2018 study using a randomized survey design assessing rape among Liberian girls incorporated modified survey design methods to improve ethical data collection relevant to the cultural and contextual contexts. This article presents the findings of a thorough review of rape scholarship and design methods. Methods: Based on a qualitative analysis of field notes by the research team, we present lessons learned and best practices identified in the planning, pilot-testing, and implementation phases of the 2018 Liberian study. Results: This study helps inform innovative design methods striving to (1) avoid using obtrusively graphic language or labels prevalent in westernized studies, (2) authentically collaborate with African experts to adapt strategies to be culturally appropriate and contextually relevant, and (3) create respectfully transparent interactions with respondents and communities. Extensive research preparation and inclusive regional expertise inform compassionate methodological techniques, yielding improved Afro-centric participant experience, low participant attrition, and quality data use in policymaking. (4) Conclusions: This article offers innovative design methods to study rape, placing context, culture, and participants at the heart. Authentic collaboration with national-level experts is vital for conducting more reliable and ethical field research in the African region.

## 1. Introduction

As 2030 approaches, there is increased attention to supporting Sustainable Development Goal 5: Gender Equity, including the mitigation of violence, exploitation, and harmful practices against women and girls. Global health research indicates that the magnitude of gender-based sexual violence (GBSV) around the world is high, especially in low-/middle-income nations, many of which are in the Majority World (with 80% or higher poverty rates) [1,2,3,4]. Extreme poverty, crises, and biological threats like COVID-19 can exacerbate rates of GBSV [5,6,7,8].

In recent decades, research unraveling the issues of GBSV, including the rape of vulnerable populations, is on the rise. However, the majority of GBV and child sexual abuse (CSA) scholarship historically pertains to studies conducted in Minority World nations (with less than 20% poverty rates), predominantly in Europe and North America, with fewer studies conducted in Majority World nations [9,10,11]. More studies are being performed in Africa. Yet, contemporary international social movements like *#MeToo* have further inspired stakeholders to prioritize conducting more studies in the Majority World [12] to improve our understanding and methods of investigation of gender-based violence [13,14,15,16].

A review of over 320 sources indicates that “research on Africa is strongly dominated by outside, non-African, mostly Western views” [17] (p. 197). A key issue is that rape surveys tend to include questions on GBSV that use foreign terms or concepts; be conducted in a manner that could be coercive, unfamiliar, or offensive for participants; and at times broach or confound taboo topics like child sexual assault and early marriage that are highly uncomfortable or triggering in specific cultural contexts [18,19,20,21,22,23]. As the body of GBV studies in the Majority World grows, the research community needs to carefully tailor our study designs to be more sensitive to the needs of participants in different parts of the world [13,24,25,26].

In the last two decades, most Majority World data on intimate partner violence and GBSV have been a product of Western-funded, nationally representative population-based surveys, with the most prominent considered to be the DHS studies. National DHS survey data are used to examine predictors or correlates of gender-based violence (GBV), as well as to inform international development policy and programming [27,28,29,30,31]. A 2022 study examining 2012–2018 data collected using the Demographic and Health Surveys (DHS) from 35 LMICs indicates that one-in-three women suffer intimate partner violence ([3], p. 465).

However, as with other Western-funded quasi-experimental research projects, there are potential limitations to DHS survey research due to flaws in its design, data variation, and methodological elements that lead to validity issues [32,33,34]. As in the case of Liberia, various population-based surveys often neglect to follow best research practices. Sexual abuse studies, including rape surveys, should incorporate “best practices, such as studying rape in the context of other health or violence questions, asking specific questions using locally relevant terms, and providing privacy to respondents, all of which lower the probability of ‘false negative’ responses” ([35], p. 451). Yet, this is not commonly realized. 

In response to the literature gap on rape in the African region, our research explores the literature and survey instruments assessing rape, mainly targeting violence against impoverished women and children. Our literature review indicates that various survey studies on sexual violence in the Majority World can be prone to methodological problems, validity issues like social desirability response bias and construct validity [36,37,38,39,40], the reliability of assessment instruments, and ethical issues of data collection in different cultural settings and vulnerable populations [25,36,38,41,42,43,44]. The purpose of this paper is to contribute to the methodological scholarship on conducting rape research in West African settings by presenting the lessons from a 2018 large-scale Liberian field survey on female statutory rape conducted collaboratively by Liberian and American researchers. 

This paper explores the literature and the case study of the 2018 Liberian research project to examine potential means of methodologically modifying and supplementing standard Westernized rape survey designs to more contextually (a) inform policy knowledge needs identified by West African experts; (b) incorporate crucial steps to promote inclusive collaboration with national experts, including West African licensed clinicians; and (c) ensure a more ethical, cultural/context-specific research design, supporting improved survey experiences for participants, many of whom may be rape survivors. After qualitatively assessing field notes by our research team, we present details of the research design implemented in Liberia, specially tailored to be culturally sensitive and contextually responsive, in the hope that it may help inform quality future rape studies in the African region. 

## 2. Materials and Methods

This paper presents the literature review and field notes of the Principal Investigator (PI) and three Co-Investigators (CIs) from the 2018 field survey entitled ‘Fatal Remedies: Child Sexual Abuse and Educational Policy in Liberia’. The mixed-method study involved a field survey of 715 randomly selected Liberian households, assessing levels of statutory rape among young women in relation to correlates like education and delayed early sexual activity that tend to mitigate rape rates. This paper first presents the synthesis of the literature review conducted during the planning and pilot testing phases of this research from September 2017 to July 2018 to inform the study design. Secondly, we present findings from a qualitative review of the field survey notes of the research team to identify key methodological strategies as well as best practices and lessons learned. 

Documentation of the research design process, pilot-testing, and feedback from experts was maintained by the PI, an American university researcher with 15 years of experience in international human rights, including eight years in Liberia. She worked in collaboration with three CIs. One CI was an American university professor with expertise in sexual assault studies, and two Liberian licensed clinicians, who both manage a social welfare nonprofit organization called Renewed Energy Serving Humanity (RESH) and lecture at the University of Liberia. Our research team met weekly to discuss the process of the study, and detailed field researcher notes were methodically taken and reviewed. Field notes are a proven technique for capturing best practices, lessons learned, and areas of concern in the research cycle to improve the quality of data collection and analyses [45,46,47,48].

During the planning phase, the PI took the lead in identifying and reviewing potential reproductive health/rape surveys prominent in the literature, with the support of the CIs. We reviewed these instruments to critique the questions and methodologies for cultural and contextual appropriateness within a West African setting like Liberia. A team of 23 volunteer African experts were invited to consult on the survey design and pilot testing phases before the data collection launched in June 2018. This localized consultant team (LCT) comprised four West African academics, four representatives from the Liberians Ministries of Gender (MOGs), five from the Ministries of Education (MOEs), two from the Liberian sexual violence unit (SVU), two police officers, five Liberian clinicians, and six nonprofits, all with critical experience in working to mitigate GBV among vulnerable child and female populations. Their feedback was summarized into the field notes each week by the PI with support by the CIs. 

Later in the pilot testing phase (May–July 2018) in Liberia, the Liberian Cis selected a team of 15 enumerators with the support of the MOG. Each data collector was from or resided in the study sites and was fluent in Liberian English. Liberia was settled by American freedmen in the 19th century, most of whom spoke Americanized English, compared to the nearly 16 indigenous languages among various ethnic societies. As English spread over time and became the national language of the country [49,50], a creolized vernacular called Liberian English developed. This version includes syntax modifications and has loan words that come from various indigenous languages, so fluency in both formal and non-formal English is important in this country. The enumerators also had extensive training and experience in survey work, social welfare, child protection, and research. The entire data collection team underwent IRB ethical training. They then supported the pilot testing of various survey designs and continually provided feedback on the study design. 

Lastly, we review the summary of the feedback captured by the Liberian adult participants from the field survey. In total, 719 households underwent the consent process. Of these, 715 households consented to participate in the survey, and 4 are documented to not have consented and thus were excluded from the data analysis. At the end of the survey, all consenting participants were asked by the enumerator whether they felt that any of the questions in the survey should or should not be asked because they felt they were uncomfortable, upsetting, harmful, too personal, or inappropriate. Overall, 91.2% agreed that the survey design was acceptable, with no significant differences between rural vs. urban participants (ꭓ^2^ = 1.11, *p* > 0.10), ethnicity (ꭓ^2^ = 10.30, *p* > 0.10), or age (ꭓ^2^ = 12.97, *p* > 0.10). Moreover, 105 of these participants provided additional qualitative feedback on their survey experience. 

There were various stages in which the feedback from the enumerators, the expert consultants, and the participants who consented to participate in the pilot testing phase was incorporated into the survey design. In the next sections of this paper, we first present pertinent issues uncovered in the literature review in the Section 1. In the Section 2, we then present a summary of the key lessons learned and best practices identified in the review of the field notes collected by the PI and three CIs, along with the research implications of this study to be considered in future research studies on rape in Majority World settings. 

## 3. Results

Our review of over 320 top-cited rape survey studies conducted indicates that the various studies that came from African settings use methodological approaches that may not be appropriate to the context and culture of their sampled populations and thus may yield misrepresentative and inaccurate statistics. In Liberia, this issue may be a major problem in our understanding of rape, especially out of the ashes of war.

Following Liberia’s civil war (1989–2003), a number of actors have attempted to raise public awareness of sexual violence and to place it at the center of post-conflict policy efforts. Local and international nongovernmental organizations, Liberian government ministries, a variety of United Nation entities, and “dozens of media reports claim that ‘75%’ (or, in some cases, ‘90% or more’) of Liberian women suffered conflict-related rape… the best available evidence suggests that these claims are likely inaccurate… [other Liberian] studies suggest that perhaps 10% to 20% of all women suffered sexual violence” ([35], pp. 447–455). Unreliable rape studies can continually misinform policy and practice in Liberia and the West African region in the mitigation of sexual violence against female children. 

Field notes were taken during the planning and pilot testing phases of the 2018 Liberian study on statutory rape, and our team consulted with regional and national experts on their feedback on the key findings from our literature review. We additionally facilitated a review of potential surveys to use in the Liberian study, including: the USAID DHS 2017 instruments; the CDC’s Violence Against Children and Youth Surveys (VACS); the ISPCAN Child Abuse Screening Tool (ICAST); the U.S. National Crime Victimization Survey; the U.S. National Intimate Partner and Sexual Violence Survey; the World Health Organization Multi Country Violence Against Women Survey (WHOVAW); the National Survey on Teen Relationships and Intimate Violence (STRIV); the Sexual Aggression and Victimisation Scale (SAV-S); the Sexual Experiences Survey—Long Form Victimisation (SES-LFV); the Short Child Maltreatment Questionnaire (SCMQ); and Safeguarding Teenage Intimate Relationships (STIR). The field notes on the feedback from the African expert consultants indicate that approximately three in four of the experts had an issue with various surveys, including the WHOVAW, STIR, VACS, and DHS instruments, capturing data on sexual abuse against female individuals. As this paper will explore, common critiques included that various GBV questions were invasive, culturally inappropriate, projected troupes of African sexuality, and were leading questions for participants. 

According to the LCT, in many contexts, including Liberia, a leading question can be defined as one that a) offers only a binary answer set (yes or no), but, moreover, offers solely one selection statement, which in turn can cause a participant to subjectively interpret that agreeing with the statement is the most correct or most desired choice on the part of the asker. Out of cultural norm, they will select this choice as to not seem offending to the enumerator, a guest whose wishes/needs should be prioritized out of a sense of social decorum, even if this means blurring the details or ‘correcting’ the guest. This cultural consideration can be a factor of truth-finding and politeness that outside researchers must account for in many settings, including among some Japanese populations [51], American Minnesotans [52], and Dominicans [53]. 

### 3.1. Regional Deficiencies in Rape Surveys in African Settings

Many of the African regional experts agreed with our finding in the literature that many sexual assault surveys predominant in the west have higher application rates of testing and information gathering in high-income nations, with less survey research employed in regions like Africa. They also agreed, after reviewing various surveys measuring rape and child sexual abuse like VACS and DHS, that they were not conducive to West African settings without modifications. 

#### 3.1.1. Diction

Firstly, standard surveys, often first developed in the Global North, use definitions of rape that evolved in the Western rape literature [54]. These definitions may not always translate across vastly different contextual and cultural settings. Legal definitions may need to be broadened to accommodate the different contexts of sexual violence in Africa ([40], pp. 55–59).

“You [Americans] uses words like ‘intimate relations’ and ‘sex’. These terms are less used in nations like here [Liberia] where people have low schooling and use colloquial terms, sometimes translated from their mother tongue [ethnic language].”—Liberian ministry official, February 2018

Secondly, the consultants offered that socio-cultural context can also impact perceptions of sexual violence as well as survivor recovery from abuse, including mental health issues [54], like post-traumatic stress disorder, depression, and stigmatization. There are limited studies on child sexual abuse in Africa because of the complexity and sensitivity around the topic. The literature notes practical and ethical obstacles to studying sexual violence in the region, including Ghana. “Successful studies [in West Africa] require investigators to carefully navigate an intricate maze of obtaining formal access and consent, involving processes meant to protect victims from further victimization during field research” ([55], p. 1). While there are various well-established survey methods in the field of domestic sexual violence, international development tends to be riddled with survey questionnaires that can be both obtrusive and triggering to a potential survivor of sexual abuse.

Our review of our team’s field notes implies that the African regional experts appeared most taken aback by questions found in the most common survey used in African settings, the USAID’s 2017 DHS7 Domestic Violence questionnaire. Specifically, they highlighted the following questions as inappropriate for Liberian contexts: 

Survey Script: “Did your last (husband/partner) …physically force you to have sexual intercourse with him when you did not want to… At any time in your life, as a child or as an adult, has anyone ever forced you in any way to have sexual intercourse or perform any other sexual acts when you did not want to? [Yes/No/Refused to answer/no answer] Who was the person who was forcing you the very first time this happened? [List of potential attackers, like current husband/partner, father/step-father, in-law, family friend, teacher, police/soldier, etc.].”

Most of the consultant team commented that the questions were leading and inappropriately obtrusive for the cultural context.

“[Such questions] will be too embarrassing [for] many women…make them feel shamed or vexed. They ask too directly, like a slap. Liberians don’t usually dialogue this way.”—a Liberian clinician examining DHS instruments in February 2018

“If I was asked these, I would look sideways, asking who is this person being so rude?”—Beninois clinician examining DHS instruments in May 2018

Three other consultants observed that this line of questioning also promoted negative troupes of sexuality among low-income populations. 

“I’m confused. Why it’s assuming women they will survey are violently abused by their husbands? It feels like more stigmatizing African men as if abusive and angry, and women are victims. Why no other option to answer for if she in relationship that is healthy or loving?”—a Liberian MOG representative examining DHS instruments in February 2018

They agreed that the wording would be perceived as foreign to most potential participants. Graphic questionnaires like these often use standardized colonizer languages in the global South, which can lead to issues of participant unfamiliarity and misunderstanding. Diction, selected by international researchers, like ‘intercourse’ or ‘sexual acts’, may not be used commonly by the population and thus often may not be fully understood by in-country participants. Using standard colonizer languages in research perpetrates neo-colonial language standards, in turn potentially invalidating indigenous, pidgin, or colloquial languages [56,57,58,59].

#### 3.1.2. Inequitable Research Team Dynamics

Furthermore, the team found fault with the fact that international development research can involve unequitable power structures among ‘northern’ researchers and ‘in-country’ researchers, in which dominant investigation decisions lie mainly with international members. Western researchers tend to promote westernized constructs of development and human rights based on international conventions. While this promotes dialogue about legal standards for the rights of all people, it can constrain localized conventional concepts of rights and abuse. Varying levels of research collaboration between foreign and in-country researchers are often linked to international networks, influences, and funding. This situation can exclude crucial national expertise and experiences [43,59,60,61,62,63,64,65]. 

“[Western agencies] come in and tell us what tool to use. They don’t ask if it is properly made, if our people will understand it or want to do it. You just assume they will. And, our research teams can’t stop the raining from falling from the sky [metaphor for the research project failing in someway].” —Beninois clinician, December 2017

Over several planning meetings in spring 2018, various African expert representatives noted the continual problems of foreign–national researchers and international agencies coming into their country to gather data from citizens, with little regard for the context in which they were researching and the impacts of rapid research efforts. 

“Many international nonprofits come in, like during Ebola, doing research on Liberians who were affected. They didn’t notify the Government [properly]. They came in with clipboards and pens, and took people’s intimate information with no consideration of compensation or the effects of their research.” —MOG representative

“If you (Western researchers) come in with your ideas of what is sexual abuse here (in West Africa), you will get wrong data. Your standards of protection aren’t universal.”—Nigerian clinician

Notably, several of the national experts that were consulted for this research acknowledge holding some level of resentment and concern about unethical practices involving data gathering in their country and the region. 

“No more cowboy researchers- coming in and taking [data] irresponsibly. They may mean well, but I read some studies that are harmful. Their data is badly collected. Not enough addressing of Liberian context or how people think and live here. International organizations want numbers, but they measure wrong, and don’t check the quality of what they gathered, how they got the information, and if it is correct.”—Liberian–Nigerian clinician

#### 3.1.3. Regional Ethical Standards

Another methodological concern of the African team was ensuring researchers applied appropriate ethical standards in the health and social sciences, which was specifically an issue in Africa. 

“Liberia has ethical standards. We have a national IRB process, but too much is done by international actors without following these standards. How do we protect our people from unethical studies if we don’t know about them.”—Liberian professor

This issue is additionally noted in the literature. Concerns include offering improved input by field staff who regularly ensure but go less-recognized for “‘doing ethics’ in the field appropriateness… [examining the] value of ethics guidelines… in Africa to ensure that local and national priorities and concerns are considered at both the micro and macro levels” ([27], p. 333). 

Additionally, a 2014 technical working group on data collection by INGOs identified numerous ethical issues in development research, including: bypassing parental consent to talk with their children about abuse; collecting data from children disclosing abuse without proper programmatic processes; a lack of evidence and consideration of what impacts may arise from interviewing children about sexual violence; and, applying different ethical research practices in LMICs than in high-income countries, such as interviewing very young children about sexual engagement [23]. The results presented in the next sub-section offer some guidelines and steps that may help improve the quality and ethical validity of future research in African settings.

### 3.2. Survey Modification Strategies for West African Settings

The African consultation team collaborated with our 2018 team of international researchers to design, pilot test, and incorporate supplemental steps in the research project design and instruments to increase the data validity by *choosing appropriate data collection strategies*. Modifying research designs that take extra precautions in data collection and partnerships can greatly reduce survey participants’ discomfort with any questions that seem intrusive, invasive, or taboo [23,35,66,67,68]. Although this field research collects sensitive material on matters of sexual practices, sexual engagement, and gender norms, the study carefully incorporates strategies recommended by recognized scholarship on rape/child protection survey design. A valuable resource recommended by various African experts included referencing the UNICEF’s Measuring Violence Against Children—Inventory and assessment of quantitative studies: 2014. The major strategies applied and tested include the following. 

#### 3.2.1. Cross-National Expert Partnerships

There is great potential in collaborating with national-level academics, government agencies, practitioners, and even research participants to further the potential of field studies to diversify interpretation, knowledge building, and more opportunities to meaningfully empower voices and sustainable change beyond simple peer-reviewed publishing. As the 2018 research tried to include, a critical step is for any foreign or Western researcher completing research in a country to have relevant experience and familiarity with the country’s culture(s), history, and the contextual reality of the in-country target population. 

Previous experience working in a country may increase a foreign researcher’s capacity to realize and interpret data collected in the field, make relevant conclusions, and recognize appropriate strategies for conducting ethical research [48,65,69]. The PI had worked previously in Liberia and the region for nearly a decade but considered her positionality. While she was aware of the context and the issue of abuse, she was cautious to learn how to conduct sensitive data collection.

“Initially, we considered doing a digital survey method for the study. But based on my experience in field [in the Hinterlands], people are less familiar with technology, and can be mistrusting of data being collected not with pen and paper, but with an iPhone. I asked ES and RN if we should do paper, though it will be more work. They said yes, for sure. They also added that if we give iPhones or ask enumerators to use them, they might be robbed, or targeted with the poor economy.” —PI notes, February 2018

“I need to find additional funding to have the tools (survey instruments) reviewed by a linguist. I know some Liberian phrases, but am not 100% on how well. ES is helping in the translation to colloquial English, but he is older. Some of our younger participants might use different terms than older Liberians.”—PI notes, March 2018

Another step is for outside/foreign researchers to seek out pertinent ways to include genuine and diverse collaboration in their studies, chiefly with national representatives throughout multiple phases of the investigation [23,54,59,70]. As indicated in this article, the 2018 study includes volunteer Liberian experts, including ones from key ministries, national universities, and nonprofits, on a consultant team throughout all phases of the research, including the data analysis, in which their feedback and ideas are prioritized as the most vital out of the research team. 

“It is important for us that we know the study followed the Government requirements, and you work with Liberians to help make the study.” —Liberian community elder

“First thing I wanted to hear is that you are IRB certified and working with others from this country.”—Liberian MOG representative

They provided ongoing feedback and reviews of the research design and its instruments throughout the entire research process.

#### 3.2.2. Validating Language Selection for Cultural and Contextual Relevance

The 2018 survey applied the legal definitions of ‘statutory rape’ to establish the magnitude of the phenomenon of female statutory rape prevalence rates. Our analysis indicates the importance of using, or at least considering, the legal definitions of rape, child sexual abuse, and their various forms, through the lens of the country in which the research population lives. Nearly all the expert consultants agreed that too often, Western legal constructs and definitions in international conventions like the Rights of the Child may clash with localized or national definitions. Additionally, they emphasized the importance of how indigenous, rural, or illiterate populations may define rape, which can also clash with legal constructs. Citizens at the community level, like parents and teachers, are less familiar with the legal constructs and, in fact, often have different perceptions of what rape is. Their definitions are influenced by the informal dominant constructs in their community, tradition/religion, or ethic group. International surveys like the DHS provide little guidance for how to modify surveys, if at all, for localized contextuality. 

“When we are saying ‘rape’ do you mean with or without consent? Do you want to measure consent of the child? If so, that is a problem, as legally children cannot consent to sex, and in many tribes, early marriage may be permitted. How will this study get around this.” —Liberian professor

“In Liberia, we say ‘man-woman-business’ but not ‘intercourse.’ Intercourse sounds like a type of road or something. A normal citizen will not use this word. The DHS survey does? Interesting…”—Liberian clinician

The 2018 study in Liberia only examined the statutory rape of girls. There are many other forms of abuse that vulnerable populations, like impoverished women and children, may face. Yet, we offer an example of how statutory rape is defined differently in Liberia. The quantitative phase assesses statutory rape using the Liberian legal parameters of early sexual engagement (sex before age 18) as being legal or illegal. 

Liberian legal precedence addresses the factors of the age of the female minor, the age of her partner or potential perpetrator, and if there was established parental consent for any sexual activity. The data reported by each female survey respondent about her sexual initiation experiences as a child or youth are used to deduce if her case meets the precedence to be tried as ‘statutory rape.’ In Liberia, statutory rape is illegal even in cases where the minor (under 18) consents and whether or not there is violence or coercion imposed [21,22,71,72,73,74,75]. The Supreme Court of Liberia states “statutory rape [in Liberia] is a strict liability crime which does not concern itself with force and resistance as necessary elements. And consent is not a defense” ([76], p. 15). Yet, most international human rights conventions indicate that statutory rape laws differ in variations of ‘age of consent’ and whether the child’s consent at the time matters and issues of force involvement [39,77,78].

This research indicates that respondents’ concepts of sexual abuse can often differ from legal definitions. Research on adolescent sexual health warns of “the blurred legal and conceptual boundaries between child marriage and sexual violence…simultaneous legality of child marriage and marital exemptions to statutory rape laws provide legal loopholes for sexual acts with children that would otherwise be considered crime” ([77], p. S72). Early sexual practices rooted in traditional ethnic customs and gender roles frequently clash with legal precedence [66,79,80,81,82]. In Liberia, it can be difficult to navigate between (a) traditional early marital practices sanctioned by the state, (b) harmful forms of early marriage that are illegal, and (c) illicit sexual activities with a minor by an adult which violate all accepted norms of early child sexual activity, placing children at an even greater risk of harm [81,82,83,84,85,86]. Section 2.2 of the Domestic Relations Law has not been repealed. It includes a provision for the early marriage of a minor between 16 and 18 years old, given that her parents/guardians consent to the arranged union [84,85,87]. Thus, we recommend that researchers consult national experts on the various formal and informal constructs of rape in relevance with international and localized definitions.

#### 3.2.3. Modifying Survey Questions

The Liberian clinicians and various ministry representatives indicated in the pilot testing phase the issue of how to inquire about rape. They agree that in the informed consent process, the purpose of the study should be made transparent, including if it involves rape and child assault. Yet, culturally, there was a large concern amongst the West African consultants about the probability of respondents experiencing psychosocial trauma from having their early sexual experiences negatively labeled as abuse. 

“It is bad to talk with small childrens about rape. It is insensitive, and they may think rape is something else or may not even know. Adults talk about it. But each people (ethnic group) may have different practices that you might say is rape, but [by Liberian law] isn’t.”—MOE official

“Liberians don’t mind talking about sex and abuse. It is just an issue of talking about their own experiences in a more sensitive way. In Africa, if you are not offering a treatment, or definitions of rape vary, it can be best to avoid using the word directly when inquiring about their own experiences. It can seem unethical [to American researchers], but the participant, if they are an adult, and are informed the study is on sexual relationships and rape are intelligent to know. But we don’t need to label it unless we have to.”—Liberian clinician

“Rape is not taboo to talk about. But it is culturally upsetting to ask a woman if she was raped, assuming her sexual history involves violence or coercion. In Liberia, if she is a girl child at time of sex, he is an adult, and the parents didn’t know, then legally that is rape. Why investigate if it was violent if it will cause bad feelings with no support later?”—Nigerian clinician

Thus, the 2018 survey decisively excludes triggering word like abuse and rape. It focuses instead on discerning if the case could be tried in a Liberian court as illegal sexual activity with a minor. The surveys were purposefully designed to use sensitive questioning techniques to mitigate potentially offending, embarrassing, harming, or shaming participants on private or socially taboo topics like sexual activity. The questioning techniques come from recommendations found in the literature review of survey resources, including 2014 UNICEF research reviewing quantitative studies involving sampling households reporting child abuse. “Research ethics [including when interviewing children or conducting sensitive lines of questioning] was seen as a challenging area that needs to be addressed” (9). 

While prior surveys used by USAID and UNICEF serve as the model, modifications are designed with Liberian expertise, piloted, and then modified again before data collection. The research informed consent procedures specify that the study is collecting data on early sexual activity and issues of sexual violence or rape among girls. Yet, the survey questions are purposefully made to avoid triggering language, like ‘rape’ and ‘abuse’, and to pair standardized English terms with Liberian colloquialisms to ensure all participants fully understand the questions. The strategy is to deduce whether a young woman’s early sexual experiences meet the legal standards of a statutory rape case based on if, as a minor, she had sex with an adult man and if her parents/guardians consented to the relationship or were aware of it prior to intercourse. The female survey is designed to first inquire about how old she was when she first became pregnant, lived with a male partner, or married to first evaluate if she likely engaged in sexual activity before 18 years of age. 

If her responses indicate that she had sex early in life, the enumerator will then ask the following: (1)“To confirm, you had sex (came into life) before you were 18 years old? (Yes/No/No response);(2)Did the man (men) have permission or consent from your parents and family to be with you? (Yes/No/No Response);(3)At the time, was the male (male) who you were with about the same age (under 18/youth) or was he an older man (18 or older/adult)? (Peer/Adult);(4)What was the occupation of the male (males) at the time they were together? (disaggregate occupation as education-based if teacher, school staff, or student, and as other if no school related).”

Based on this line of questioning, the research deduces the categorization for each female case. Most of the surveys purposely include questions that provide data that are used *to deduce* sexual engagement, risk levels, and rape without specifically using culturally abrasive word like *rape* or *abuse*. This step is recommended by Liberian mental health experts as a means of protecting participants from unnecessarily harm such as the psychological trauma of naming prior experiences. 

“It is more appropriate to not label these women’s experiences, make them shamed. They know likely if there was violence or they were forced. But it is not culturally sensitive to label an experience, especially as a stranger, if it cannot help the situation. It is better to ask in a way that is more sensitive.”—Liberian clinician

“An American clinician might say use the word rape, but this is not always best in Africa. It can shut people down to hear it. It’s abrasive, too blunt, and ugly. Rape has many connotations and denotations. It has a long history especially with the war.”—Liberian sociologist

In the 2018 study, the feedback on using the word rape and asking about violent sexual experiences was almost unanimously shot down by the Liberian clinicians and most of the social worker enumerators. Secondly, no treatment (like therapy or PSS programming) is offered to participants by this study; thus, the researcher and consultant team conclude that it could be unnecessarily triggering to guide female participants through invasive questioning. During the initial phases of the research project, the LCT and data collectors provided their perspectives concerning the issue of this line of questioning about individual rape experiences. Approximately four out of five members raised concerns about asking explicit questions on rape. Our decision to minimize invasive questioning techniques was later validated when an initial analysis of the statistical data in the 2018 study significantly captured reliable and valid measurements of rape, despite negating the use of graphic rape questions. 

“In the meeting with our enumerators, G.I. shared that the questions (listed above) were good. She tried them out pilot testing last week on several female colleagues, and a few neighbors (their data was not included in the study). Margaret, Prisie, and others found the same. They said not to ask about if the sex was violent. I am still wondering about this, but will go with their feedback.”—PI notes, June 2018

For this reason, we posit that research does not always need to rely on the inclusion of direct questions using words like “rape” to ensure quality data collection. Researchers must carefully reflect on the impact of conducting emotionally charged research on personal topics like violence and rape in social science, particularly weighing the potential costs to their subjects, who may be asked to relive their trauma in detail. Investigators should carefully consider their relationship to their research subjects and the impact of the methods and language used in investigating their intimate experiences [9,14].

#### 3.2.4. Pre-Survey Community Awareness and Approval Processes

As highlighted earlier, another tested data collection strategy that the research incorporates is to raise awareness of the research goals and introduce its instruments and enumerator team to community leaders. The goal is to gain local buy-in before any data collection. Informed community leaders may then choose to spread the word about the study to residents as a means of informally approving the research.

“Liberians will culturally feel it needed to talk with a stranger who come unexpected, especially someone who seems official, if they come to their door. But, if you go to the community elders first, explain the process and purpose of the study, and give them the tools (informed consent script and surveys), they will understand their people don’t have to participate unless they want to. Deh will tell their community households about the project, and say that they can choose to participate, and the importance of not doing so so if not comfortable. Maybe they [the participants] can choose not to open the door at the knock, or simply say no. It is best to tell them who is coming, for what purposes, and what days and times so they can be informed and not surprised badly.”—Liberian professor of sociology

“Unlike in England or US, the people here (Liberians) look to their community leaders for guidance on participation. They trust them, but don’t trust outsiders. You need to be invited by the community leaders or else you may face upsetting people.”—Liberian clinician

“You go into a community without permission and expect thunder (upsetting people)”—Nigerian researcher

“You must follow traditional practices, and ask permission of those in charge, or you violate rights and risk bad resentment. The elders know how to tell their people of a study, and they work with nonprofits and researchers a lot to do so in an ethical way. No one feels forced but they understand by explaining it additionally by the leader’s words the consent process”—Liberian social worker

Community leadership buy-in is vital in many West African settings, where strong ethic/tribal systems make up a large portion of the community ecology. During the planning phase, representatives from all the proposed survey sites are asked to gather to review documents, meet with the research leadership team, and, after careful consideration, grant permission to proceed. It may not be necessary to follow this type of community engagement strategy in other LMIC settings, yet it is important to realize that in many parts of West Africa, community leaders hold the responsibility of protecting the interests of their people, and many Africans turn to community leaders for guidance when it comes to research participation and knowing if they can trust enumerators who are outsiders. A Liberian clinician stated, “In Africa, we do things in communities a certain way. You need to follow those ways or you won’t get too far.” Safeguarding community-based participatory research and context-specific collaborative research approaches, particularly in early research phases, can improve data quality and lower threats to validity [70,88,89,90,91].

#### 3.2.5. Triple Informed Consent Procedure

This research further includes a thorough informed consent procedure for all respondents. The young woman’s survey instrument contains a three-phased consent process. The purpose is to ensure each female respondent verbally consents to being willing and comfortable to participate at the start, during, and at the end of the survey experience.

“I like that [the enumerator] asked me if I want to keep going with the questions in the middle of the talk. It make me feel like they care and there no force”—Female Liberian participant, 34 years-old, during pilot

“It’s nice, the [consent] process. The data collector was showing me way to stop. I feel easy, safe. I liked it [being asked three times]”—Liberian participant, 22, data collection

“At the end of the questioning, maybe a person wants to stop. They don’t know if that is okay since they say yes [to participating] at the start”—Liberian social worker

The process also provides detailed instructions for withdrawing her responses at a later date, if so desired. The feedback from participants during the pilot testing indicates that having the three-phased consent makes potential participants feel more secure in their understanding of the research and their rights as participants, as well as creates a general feeling that their feelings and experiences are valued by the research team.

#### 3.2.6. Building Survey Familiarity and Transparency

In the last weeks of planning, five representatives of the expert consultant team were concerned with making participants aware of the line of questioning before they shared their own personal experiences of early sexual engagement.

“There are not many studies on rape here. Those done where after the war, when Taylor in power, and so rape is seemingly a war thing. It is okay to ask Liberian women about their experiences, but it is important to help them know the questions that they will be asked”—Liberian clinician

“People may talk about sexual violence, but being asked about it can be a new experience. They may not understand until they experience the full survey”—Liberian SVU police officer

Thus, we worked with the African clinicians to design a research strategy that is a novel technique, termed Answering for Your Confidant, as a way to introduce a participant to the types of sensitive questions before she is asked to share details on her own experiences. Secondly, the technique aims to build her awareness and context of the answer options for each question and, lastly, to provide time for her to consider responding about her own private life. Approved by all the clinicians on the national consultant team as appropriate for an African context, the goal is to use this innovative approach at the start of the survey. The enumerator asks each young woman participant to first practice answering specific questions about the sexual engagement history of an unnamed, close female confidant about her same age. This person may be a sister, cousin, or best friend who she confides in regularly. It is recommended to exclude these data from the data analysis. The purpose is to simply introduce the participant to the experience of answering the question set.

The enumerator guides the respondent to answer the series of sensitive questions for her confidant (without ever naming her), including at what age she started having sex, became pregnant, or married. After the ‘Confidant Section’ is completed, the data collector then asks if the female participant herself would be comfortable answering the same questions and reporting her own sexual experiences. This strategy design was inspired by the researcher’s own experiences talking about her sexual abuse with mental health professionals. When talking with other survivors of abuse, she realized that many of them share the desire to first be made aware of what exactly might be asked to minimize the traumatic experience of broaching sensitive memories without some level of preparation, which can feel like getting on a rollercoaster without adequate seat restraints.

Applying the ‘Confidant’ technique seemingly improved the general participant response rate during the study’s pilot. Additionally, this step is approved by field experts for its potential value in lowering self-reporting bias as well as improving ethical transparency for participants. To date, this technique appears to be entirely new in rape survey research and may be considered in future sexual abuse studies.

Of nearly the 105 participants who were asked by enumerators about their comfort level with taking the survey during the implementation phase, nearly all provided positive feedback. In general, the enumerator team found this strategy to be a positive addition to the research, although none of the close confidants’ information was incorporated into the final analysis but instead deleted from the quantitative analysis, as it was second-hand sourced with no ethical permissions to be included. The survey questions in relation to this section were carefully reviewed and approved by both the University of Pittsburgh and the Liberian IRB process.

## 4. Discussion

Both qualitative and quantitative research on gender-based sexual violence (GBSV), as well as evaluations on rape-prevention programming, have continued to increase since the 1990s [14,15,16,92]. The review of sexual assault and sexual violence studies indicates that most research focuses on populations within the Minority World [16,93,94], yet there is a growing body of research conducted in low-/middle-income (LMIC) nations in regions like Africa, which improves the understanding of this global issue [4,93,94,95,96]. Yet, most survey designs, practices, and instruments are tailored for Westernized populations by experts from the Global North, with fewer recognized designs specifically tailored for the Global South and/or created or co-designed by Southern experts [17,25]. In this paper, we refer to ‘Southern experts’ as professionals from Majority World nations with university or vocational training and work experience relevant to the subject at hand, a common terminology in international development scholarship [97,98,99]. While this paper concerns the African region, some of our findings and conclusions may extend to other lower-income regions like Latin America, the Middle East and North Africa (MENA), and South–East Asia. Additionally, there are key research gaps that can lead to biases and misunderstandings of sexual violence experienced by vulnerable populations in LMIC settings, especially involving vulnerable populations from conflict and poverty. Additionally, sexual violence scholarship must be concerned about the accuracy and reliability of collected survey data in many low-income, diverse settings [7,8,13,95,96,100,101,102].

Researchers and practitioners need to re-evaluate contemporary methods typically employed to study sexual violence, especially in international development and global health sciences. When sexual abuse research incorporates design techniques and questionnaire/survey instruments that are not conducive to a culture and context, it can have devastating effects in terms of both lowering the data quality and having negative implications for research participants [25,26,100,102]. Additionally, health researchers have an ethical responsibility to avoid using methods and instruments that can be triggering for survivors and their loved ones, especially in low-resourced environments where vital social welfare and mental health programs are not readily available to study participants [42,43,100].

Additionally, the diction within surveys can have huge ramifications. There are competing definitions of sexual abuse in the African rape literature. For instance, a study by Cohen and Green (2012) analyzes “four systematically sampled survey investigations from postwar Liberia, no two of which used the same definition of sexual violence” ([35], p. 456). Constructs of rape and sexual abuse can influence statistics. Additionally, many legal constructs of various forms of rape can differ among countries, international conventions, and, at the decentralized level, within individual communities [35,39,40,77,78].

The review of the 2018 Liberian study case offers some unique and innovative methods for approaching measuring rape in a more culturally responsive and contextually relevant manner, maximizing ethical rigor, and ways to adjust research design with equitable inputs from African experts. We present key methodological steps that researchers can incorporate into their survey research in each phase (see Figure 1). The generally positive feedback on the survey design from our expert consultant team and over 700 survey respondents and the double ethical approval of the study design help indicate that the study meets ethical standards and places the comfort of survey participants and their cultural context at the heart of the research design.

Our paper presents those strategies that our research team specifically chose to incorporate with support from the LCT. Yet, there is a range of other potential strategies or approaches that researchers should consider choosing amongst to find the most suitable methodologies for their individual projects. Not all the steps presented in Figure 1 may be relevant to every study. The selection of alternatives should be carefully researched and assessed. Our design included making participants more familiar with the questions that we were going to ask them during the survey by having them answer them for a confidant first, and we excluded the information gained from these responses. Yet, in many contexts, this alternative may not be appropriate, so it is best to consult both the national IRB committee and the LCT. Additional research alternatives not mentioned can include

Researching and weighing the various definitions of violence in international conventions, national laws/policies, and constructs in informal socio-political institutions such as ethnic community justice systems [23];Considering whether to survey or interview vulnerable populations like children (17-years-old or younger) and elderly people (65 and older) [23,103,104];Incorporating the method of authority—prioritizing knowledge acquisition from the knowledge and wisdom of prominent people recognized within their society as having a better insight into their environment than ordinary people (e.g., children, people with a lower social standing), such as elders, religious heads, or chieftains [105,106,107,108];Considering the importance of the mystical method; many African societies consider the accuracy of knowledge to not lie with an ordinary person but to reside within a supernatural source who is viewed as an authority of knowledge production, such as a zoe (traditional healers), Sande/Poro leadership, or a religious head [105,109,110];Considering if the research would benefit more from being led by only host-national investigators in lieu of partnering with foreign researchers; asking if the foreigner has enough experience to effectively contribute; and relying on national experts to determine if a collaboration is advised [97,98].

The methodological strategies shared in this paper may not fit in every space and time. Some may require additional testing and refinement. Added steps like extending the pre-implementation consultation to various stakeholders may also require further resourcing, funding, and time. But we posit that the costs can result in a more rigorous, inclusive, and ethical design. Moreover, this study highlights the potential value of avoiding graphic survey questions in a global South environment where cultural norms limit familiarity and comfort when speaking about sexual violence. Instead, the study indicates the potential value of avoiding asking invasive questions about potential survivors’ sexual experiences, avoiding the negative labeling associated with them, and instead deducing rape status based on legal parameters (so long as the consent process explicitly states the purpose of the study examining issues of GBSV and rape). A major research design concern in the 2018 study is ensuring transparency and facilitating a trusting interaction with participants during data collection to try to eliminate any potentially harmful or obtrusive experiences during their survey process. Rape survey techniques prominent in Westernized research may not be fully suited to certain LMIC settings [35,43,95,96,100,101].

## 5. Conclusions

In global health, we are continually assessing the methods by which to ethically obtain data on GVSV and rape. Yet, we recognize the limitations of these techniques not undergoing a cross-national assessment. Future research can test the relevance of these techniques in different nations. Additionally, some of the supplemental design modifications may clash with ethical standards held by some Global North researchers. When this paper was first presented in a 2021 conference, several American sociologists in the audience critiqued the ethical strategies that were tested and applied. The response from our team was to highlight that our study involved robust inputs from (a) seven licensed clinicians, including several who had worked with rape survivors for decades; (b) two IRB ethical approvals; and (c) regular feedback from 23 expert consultants and 15 social workers enumerators. Their Africanism does not diminish their level of expertise on child protection and GBSV or make their methodological contributions any less ethical. But incorporating African expertise instead contributes to the relevance of this type of research in LMICs. Furthermore, the survey received a 99.2% approval rating among adult participants.

We urge researchers to carefully question their positionality and potential biases against alternative methods presented by Southern clinical and academic researchers. Our community must be cautious to not prioritize accepting strategies developed by Northern scholars and clinicians while continually discounting new or established strategies developed by Southern experts solely because they may seem unfamiliar or irrelevant in Western settings. “You [should be careful to] question the validity of this study, but we followed all the guidelines. Maybe it’s difficult for you to understand, as you come at this with the eyes of a foreigner where the practices in your field may take one form or another, but would not fly in my country. Who is best positioned to know what works in my home if not us [African experts],” one of our CIs noted in his conference notes. Standardized Westernized rape survey methods do not always hold up in new settings. At a time when more GBV studies are being conducted around the world, it is imperative for researchers and practitioners to reconsider the methods that we typically employ and who gets to play an equitable role in determining how best to conduct good research within nations like Liberia. Our field may need to gradually shift how we approach conducting sensitive rape research in new environments. While our intentions may be genuine, if culture and context is not placed at the heart of the research design, the result may lead to negative or harmful survey experiences for the very people who are our target beneficiaries—rape survivors.

## Figures and Tables

**Figure 1 healthcare-11-03053-f001:**
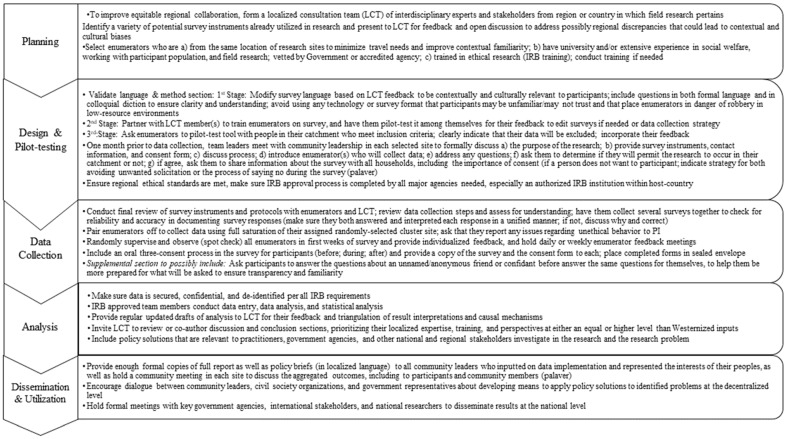
Methodological framework for contextually relevant and culturally sensitive survey research in Majority World settings.

## Data Availability

The data presented in this study are available on request from the corresponding author. The data are not publicly available due to privacy and ethical concerns.

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
