# Peer review of "Conducting Ethical Field Research on Rape in West African Settings: Case Study of 2018 Liberian Field Survey"

_healthcare, 2023, doi:10.3390/healthcare11233053_

Round 1

Reviewer 1 Report

Comments and Suggestions for Authors

The study presents a significant advancement in the field of rape scholarship in West Africa by highlighting the limitations of Westernized approaches and emphasizing the need for culturally and contextually relevant research methods. However, while the study excellently outlines the current issues in this area, it could benefit from further elaboration on alternative solutions and the practical application of the rich data gathered.

-          Strengths in Addressing Current Problems:

The study appropriately underscores the drawbacks of using Westernized approaches in researching sensitive topics like rape in West Africa. It effectively emphasizes the importance of culturally adapted survey design methods for ethical data collection within cultural and contextual contexts.

-          Need for Elaboration on Alternative Solutions:

The study could further enhance its contribution by elaborating more on alternative methodologies or approaches. While it identifies the need for culturally adapted strategies, it would be beneficial to offer more specific examples or models of these methods. Exploring and detailing other potential strategies or considering contrasting approaches would enrich the discussion and help researchers choose the most suitable methodologies.

-          Utilization of Rich Data for Methodological Development:

The study mentions a qualitative analysis of field notes but doesn't explicitly mention utilizing the rich data gathered in developing a clear model or methodological protocol. Utilizing this data could aid in the development of a more structured and replicable methodology for future studies. Providing insights into how the lessons learned and best practices identified from the data were translated into refined research protocols or models would strengthen the practical implications of the study.

In essence, while the study effectively highlights the need for culturally sensitive research methods, it could further bolster its impact by delving deeper into alternative solutions and demonstrating the application of the data collected in developing robust methodologies for studying sensitive topics like rape in West Africa.

Author Response

Dear Reviewer,

Comment 1: “The study presents a significant advancement in the field of rape scholarship in West Africa by highlighting the limitations of Westernized approaches and emphasizing the need for culturally and contextually relevant research methods. However, while the study excellently outlines the current issues in this area, it could benefit from further elaboration on alternative solutions and the practical application of the rich data gathered.”

Response: Thank you for this. This was what we are hoping is the key takeaway.

Comment 2: Strengths in Addressing Current Problems: The study appropriately underscores the drawbacks of using Westernized approaches in researching sensitive topics like rape in West Africa. It effectively emphasizes the importance of culturally adapted survey design methods for ethical data collection within cultural and contextual contexts.

Response: Thank you.

Comment 3: Need for Elaboration on Alternative Solutions: The study could further enhance its contribution by elaborating more on alternative methodologies or approaches. While it identifies the need for culturally adapted strategies, it would be beneficial to offer more specific examples or models of these methods. Exploring and detailing other potential strategies or considering contrasting approaches would enrich the discussion and help researchers choose the most suitable methodologies.

Response: This paper focuses on presenting the strategies that we both developed and pulled from key resources. Based on this recommendation, and that below, we developed Figure 1 to help highlight alternative methods at each stage in the research process. (For more details, please see the revised manuscript)

Comment 4: Utilization of Rich Data for Methodological Development: The study mentions a qualitative analysis of field notes but doesn't explicitly mention utilizing the rich data gathered in developing a clear model or methodological protocol. Utilizing this data could aid in the development of a more structured and replicable methodology for future studies. Providing insights into how the lessons learned and best practices identified from the data were translated into refined research protocols or models would strengthen the practical implications of the study. In essence, while the study effectively highlights the need for culturally sensitive research methods, it could further bolster its impact by delving deeper into alternative solutions and demonstrating the application of the data collected in developing robust methodologies for studying sensitive topics like rape in West Africa.

Response: We added the following: Second paragraph below Figure 1: “Our paper presents those strategies that our research team specifically chose to incorporate with support from the LCT. Yet, there is a cadre of other potential strategies or approaches that researchers should consider in choosing amongst to find the most suitable methodologies for their individual projects. Not all the steps presented in Figure 1 may be relevant to every study. Selection of alternatives should be carefully researched and assessed. Our design included making participants more familiar with the questions that we were to ask them during the survey by having them answer them for a confidant first and excluded the information gained from these responses. Yet in many contexts, this alternative may not be appropriate, so it is best to consult both the national IRB committee and LCT. Additional research alternatives not mentioned can include:

  • Researching and weighing the various definitions of violence in international conventions, national laws/policies, and constructs in informal socio-political institutions such as among ethnic community justice systems [23]
  • Considering whether to survey or interview vulnerable populations like children (17 years-old or younger) and elderly peoples (65 and older)[23,95,96]
  • Incorporate method of authority- prioritizing knowledge acquisition from the knowledge and wisdom of prominent people recognized within their society as having better insight of their environment than ordinary people (e.g., children, people with lower social standing), such as elders, religious heads, or chieftains[97–100]
  • Consider the importance of the mystical method; many African societies consider the accuracy of knowledge to not lay with an ordinary person but to reside within a supernatural source, who are viewed as authorities of knowledge production, such as zoe (traditional healers), Sande/Poro leadership, or religious head [97,101,102]”

Reviewer 2 Report

Comments and Suggestions for Authors

Manuscript ID: healthcare-2718689

Type: Article

Title: Conducting Ethical Field Research on Rape in West African Settings: case study of 2018 Liberian field survey

1.  Line 68.  I am not sure if this is the journal's method of citing pages (p451).  Some journals prefer "p. 451" or [35: 451].

2.  Line 160.  Did you mean to start a new paragraph or a continuation of the previous paragraph?

3.  Line 219.  Some definition of "leading" is needed as I don't see the questions as leading but perhaps extreme in the use of "ever" or "anyone" but is that leading the participant to answer yes (or no)?

Author Response

Dear Reviewer,

  1. Line 68.  I am not sure if this is the journal's method of citing pages (p451).  Some journals prefer "p. 451" or [35: 451].

Response: We used the format that the Journal requests, and incorporated all citations using Mendley using this format; thank you for double-checking

  1. Line 160.  Did you mean to start a new paragraph or a continuation of the previous paragraph?

 Response: Great catch; we did not, and have adjusted it. Thank you!

  1. Line 219.  Some definition of "leading" is needed as I don't see the questions as leading but perhaps extreme in the use of "ever" or "anyone" but is that leading the participant to answer yes (or no)?

Response: Thank you so much- this is very valuable advice. We added: “According to the LCT, in many contexts, including in Liberia, a leading question can be defined as one that a) offers only a binary answer set (yes; no); but moreover, offers solely one selection statement, which in turn can cause a participant to subjectively interpret that agreeing with the statement is the most correct or most desired choice on part of the asker. Out of cultural norm, they will select this choice as to not seem offending to the enumerator, a guest whose wishes/needs should be prioritized out of a sense of social decorum, even if this means blurring the details or ‘correcting’ the guest. This cultural consideration can be a factor of truth-finding and politeness that outside researchers must account for in many settings, including among some Japanese populations [49], American Minnesotans [50], and Dominicans [51].” (line 183 start)

Reviewer 3 Report

Comments and Suggestions for Authors

This is a well-written paper, reaching important conclusions for research on sexual assault, sexuality (and other issues) in "non-Western" cultures such as those in Africa. The points made about the non-relevance of many American terms related to "sexual assault" are well made, and this paper reaches valuable conclusions. The qualitative methodology in reaching these conclusions is acceptable.

A few minor questions: is "Liberian English" the same as Krio language of Sierra Leone (an English-African language resulting from the blending of English and African languages during, and post-slavery?). Does Liberia have a history of settlement of repatriated slaves, similar to Sierra Leone, a country I have some experience of? 

The quotations in the article need editorial attention, in terms of italicization, indentation etc.  Easily done.

Author Response

Dear Reviewer,

This is a well-written paper, reaching important conclusions for research on sexual assault, sexuality (and other issues) in "non-Western" cultures such as those in Africa. The points made about the non-relevance of many American terms related to "sexual assault" are well made, and this paper reaches valuable conclusions. The qualitative methodology in reaching these conclusions is acceptable.

Comment 1: A few minor questions: is "Liberian English" the same as Krio language of Sierra Leone (an English-African language resulting from the blending of English and African languages during, and post-slavery?). Does Liberia have a history of settlement of repatriated slaves, similar to Sierra Leone, a country I have some experience of? 

Response: Good questions: We have added the following:

“Liberia was settled by American freedmen in the 19th century, most of whom spoke Americanized English, compared to the nearly 16 indigenous languages among various ethnic societies. As English spread over time and became the national language of the country [49,50], a creolized vernacular called Liberian-English developed. This version includes syntax modifications and has loan-words that come from various indigenous languages, so fluency in both formal and non-formal English is important in country.” (about line 128)

Comment 2: The quotations in the article need editorial attention, in terms of italicization, indentation etc.  Easily done.

Response: Thank you- these changes have been made per the examples found in the journal.